# Identifying Risky Vendors in Cryptocurrency P2P Marketplaces

## ABSTRACT

Peer-to-Peer (P2P) cryptocurrency exchanges are two-sided marketplaces, similar to eBay/Craigslist, where individuals can offer to sell cryptocurrency assets in exchange for payment. Due to disintermediation, these marketplaces trade off increased privacy for higher risk (e.g., scams/fraud). Although these marketplaces use feedback systems to encourage healthier transactions, anecdotal evidence suggests that feedback often fails to capture vendor-associated risks. This work is the first to document the online safety of cryptocurrency P2P marketplaces, identify underlying issues in feedback-based reputation systems, and propose improved mechanisms for predicting and monitoring risky accounts. We collect data from two cryptocurrency marketplaces, Paxful and LocalCoinSwap (LCS) for 12 months (06/2022–06/2023). The data includes over 396 000 listings, 67 000 vendors, and 4.7 million historical feedback for Paxful; and about 52 000 listings, 14 000 users, and 146 000 feedback for LCS. First, our empirical data shows that the current feedback system does not sufficiently convey enough information about risky vendors, and is susceptible to reputation manipulation through user collusion and automation. Second, combining various publicly available information, we build machine learning models to predict account suspension, and achieve a 0.86 F1-score and 0.93 AUC for Paxful. Third, while our models appear to have limited transferability properties across markets, we identify which features most help account suspension across platforms. Finally, we perform a month-long online evaluation and show that our models are significantly more successful than mere feedback-based reputation schemes at predicting which users will be suspended in the future.

## CCS CONCEPTS

• **Security and privacy** → **Social aspects of security and privacy**; • **General and reference** → **Measurement**.

## KEYWORDS

Online marketplace, Reputation system, Cryptocurrency, Financial fraud, Sybil attack, Online safety and trust

**ACM Reference Format:**
Anonymous Author(s). . Identifying Risky Vendors in Cryptocurrency P2P Marketplaces. In *Proceedings of In submission to ACM Web Conference 2024.* ACM, New York, NY, USA, 12 pages. https://doi.org/10.1145/nnnnnnn.nnnnnnn

## 1 INTRODUCTION

A P2P cryptocurrency exchange is a two-sided market that facilitates trade between buyers and sellers of cryptocurrency assets.

*Vendors* post advertisements to buy/sell cryptocurrencies through various payment channels, such as bank transfers, gift cards, and mobile payments. *Customers* browse the market and respond to ads to initiate transactions. These platforms are different from centralized cryptocurrency exchanges such as Binance or Coinbase, where the platform matches buyer and sellers through an order book. Rather, P2P cryptocurrency marketplaces more closely resemble marketplaces such as eBay, Craigslist, and online anonymous marketplaces [11, 14, 39]. Similar to those online marketplaces, there exist malicious actors who attempt to defraud users. They may reverse payments, send fake/manipulated gift card receipts, harass users to release payments or block the release of cryptocurrencies. For the platform to be trustworthy, users should thus be provided with signals that allow them to assess counterparty risk. Most platforms use feedback-based reputation systems, where customers give vendors feedback to assess the vendors' credibility; however these systems are susceptible to various types of attacks and manipulation such as self-promoting, whitewashing, retaliation, and bad-mouthing [7, 21]. To evaluate the current feedback-based reputation system, we collect data from two leading P2P marketplaces—Paxful [2] and LocalCoinSwap (LCS, [1])—over 12 months, and monitor user activity including profile changes, posted advertisements, feedback received, and account suspensions.

We first argue that merely looking at reputation scores (i.e., negative reviews) is insufficient to identify risky accounts. In particular, 1) we manually classify negative reviews into several categories, and investigate what types of information they convey and how reliable they are; 2) we show empirical evidence of self-promoting attacks and identify the features that illustrate user collusion and automation (e.g., feedback rate, feedback interval). Next, to better identify risky vendors, we test seven machine learning (ML) models to predict account suspension. Besides reputation metrics, we combine different publicly available information such as user profiles, ads, and trade information, thereby obtaining a more precise representation of suspicious accounts. We perform the same experiment on LocalCoinSwap (LCS) and test the generalizability and transferability when using attributes common to both platforms. We then conduct an online experiment to evaluate the practical usefulness of our model. Rather than immediately suspending accounts—and risking false positives—we attempt to improve the moderation process by prioritizing accounts identified as "risky." We prepare three sets of accounts: 1) users with a high likelihood of suspension from our ML model, 2) users with the lowest reputation scores, as well as 3) a baseline consisting of a random user sample. We compare these three groups over a month in terms of the suspension rates and trading count. Our main contributions are as follows.

(1) Our study is the first to evaluate online safety and trust in cryptocurrency P2P markets by creating year-long datasets and formalizing the methodology of data collection.
(2) We empirically show the limitations of feedback-based reputation by manually investigating review quality and finding evidence of user collusion and automation.

(3) We develop a mechanism to identify account suspension using only public signals, and achieve a 0.86 F1-score and 0.93 AUC using tree-based ensemble approaches in one of the largest cryptocurrency P2P markets.

(4) While our model itself has limited transferability across P2P platforms, we distill features critical to both platforms.

(5) Our online evaluation illustrates that our model is significantly better at proactively identifying risky accounts than existing reputation systems.

Our method can help platforms design safe and more secure environments and could help moderate suspicious activity potentially more efficiently. Our findings could also improve user experience by allowing users to more accurately identify (un)trustworthy vendors. Last, given the overall scarcity of empirical research on reputation systems due to data availability, our work benefits not only cryptocurrency P2P exchanges but also other online marketplaces to design more informative reputation systems, as a complement to existing feedback-based systems.

## 2 BACKGROUND

This section overviews cryptocurrency P2P marketplaces, describes transaction mechanisms, and delves into the role of the reputation system, its vulnerabilities, and possible attacks.

### 2.1 P2P cryptocurrency marketplaces

By providing lower friction than alternatives, Bitcoin [27] and other cryptocurrencies have been used for international remittances [48]. Also, despite being far from anonymous (with a few exceptions like Monero or Zcash), modern cryptocurrencies provide stronger privacy than most other electronic payments, and have been used in online anonymous marketplaces [11, 39], for malware and extortion payments [13], or even financial scams [17]. Additionally, due to their high volatility [53], related financial products, e.g., derivatives [40], have become increasingly popular as a speculative instrument.

While most people trade cash for cryptocurrencies through large centralized exchanges (brokerage or order-book style) such as Coinbase or Binance, cryptocurrency peer-to-peer (P2P) exchanges became popular by improving privacy through disintermediation. Anecdotally, those exchanges attract customers from emerging countries in Africa (Kenya, Nigeria, Ghana), Asia (China, India, Pakistan, Philippines, and Vietnam), and South America (Argentina, Columbia) where economic and/or political circumstances may limit available financial operations [42]. For instance, in Paxful, gift cards appear to be often used for remittances from the USA to Nigeria [3]. As such, P2P cryptocurrency exchanges are a plausible alternative for those with limited access to financial services.

P2P exchange mechanisms differ from centralized exchanges and are akin to other online marketplaces such as eBay, Craigslist, or Facebook marketplace. *Vendors* set offer prices for cryptocurrency (Bitcoin, Tether, etc) and post advertisements, indicating whether they want to buy or sell. Advertisements include payment type (e.g., bank transfer, mobile payment, gift cards), fiat currencies (e.g., USD, EUR, KES), and possible ID requirements for customers. *Customers* visit the exchange website, search for ads, and initiate transactions while communicating with vendors. P2P exchanges originally focused on face-to-face transactions; but a small portion

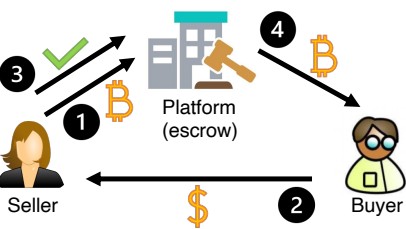

**Figure 1: Transaction flow in cryptocurrency P2P markets.**

of exchanges (e.g., LocalMonero) still offer this option, and face-to-face transactions represent a low percentage of all activities We thus only focus on online transactions.

We distinguish between custodial and non-custodial P2P exchanges. In *custodial* exchanges—such as Paxful—to initiate a transaction, users need to first send their cryptocurrency to the exchanges' wallet. *Non-custodial* exchanges like LocalCoinSwap (LCS) allow users to keep full control over their funds, and to directly exchange cryptocurrency between user wallets. In both cases, the platform acts as an escrow agent and moderates user disputes.

Figure 1 highlights the process for a transaction between a vendor (seller, here) and a customer (buyer, here): ❶ The seller sends or locks cryptocurrency (e.g., Bitcoin) to an escrow account from their wallet (either self-hosted or on the platform). ❷ The buyer pays the seller using a bank transfer, gift card, or other form of payment. ❸ The seller confirms the payment and notifies the platform. ❹ The platform releases the cryptocurrency to the buyer.

In addition, Know Your Customer (KYC) requirements may exist depending on the exchange and circumstances. For example, Paxful asks users to immediately complete identity verification if they are in a listed country [33]; otherwise, identification is required when transaction volumes exceed a certain threshold, e.g., 1 000 USD. On the other hand, in LCS, ID verification is optional.

### 2.2 Reputation systems: benefits and challenges

Since the dawn of the internet, online marketplaces have become the *de facto* place to exchange goods and services and help reduce inventories [41]. Without face-to-face communication, however, users face the risks of not seeing actual products, being cheated, or dealing with malicious vendors. Most exchanges build a reputation system to advise users on vendor credibility [37]. These systems are reportedly more accurate than word-of-mouth [38], and more effective at disseminating information. A number of studies discuss the role of online reputation and how it leads to safer and more efficient online communication, e.g., by looking at reputation system design [28, 41] or reputation impact on product price [25, 38].

Despite these benefits, reputation is vulnerable to manipulation such as whitewashing (re-entering the market under a different identity after having engaged in questionable transactions), Sybil attacks (fake accounts operated by a unique entity), slander, retaliation, and bad-mouthing [7, 21, 22]. In this paper, we focus on *self-promoting attacks*, which Hoffman et al. [21] defines as "attackers seek[ing] to falsely augment their own reputation," by submitting fake positive feedback about themselves through their *own* Sybil

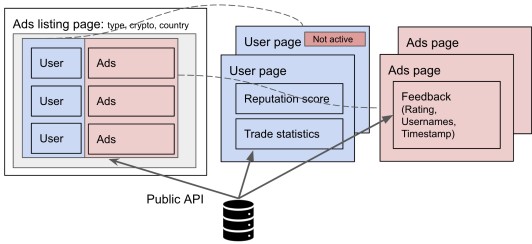

**Figure 2: Paxful webpage architecture and data collection through APIs**

accounts. Platforms that do not require user authentication or proof of interaction (e.g., payment) for feedback are particularly vulnerable. Self-promoting attacks can be conducted by a single entity or by colluding entities. We observe evidence of such attacks in our data, as we discuss in §4.2. Unsurprisingly, empirical evidence suggests the existence of SRE (seller-reputation-escalation) services to perform self-promoting attacks in online marketplaces [50].

## 3 DATA

This section describes how we collect data and identify account suspension.

### 3.1 Collection

We collect data through Paxful's publicly available APIs from June 8, 2022 through June 26, 2023. On April 4, 2023, Paxful announced it suspended operations. Although operations eventually resumed a month later, data posterior to April 4, 2023 present oddities, including a large-scale account ban, so we choose to exclude them from account suspension prediction.

We query listed ads approximately every 100 seconds. The query API requires us to specify trade types (SELL or BUY), types of cryptocurrencies (BTC, USDT, USDC, or ETH), and a list of countries. Because measuring every single country would be impractical, based on the number of transactions we historically observed on LocalBitcoins, we choose to limit ourselves to 10 countries (Russia, US, UK, Nigeria, Columbia, Germany, India, Peru, Kenya, and China) and the "Worldwide" option. Thus, we may miss ads *only* posted in other countries. (Most of the ads are cross-listed in multiple countries.) Ads include information on the type of cryptocurrencies sought or offered, fiat currencies sought or offered, payment methods, price (and its deviation from the market price), and customer ID requirements. We visit all vendors with active ads at least once a day to longitudinally record their profile and activity statistics, which will be used to evaluate our risk profiling methods in §7. We also collect historical feedback left by a customer associated with each ad as a one-time data collection. Feedback includes a textual review, rating, creation time, and the handle of the user giving feedback. Based on feedback data, we construct a "feedback graph" where each node represents a user and an edge $A \rightarrow B$ represents feedback from customer $A$ to vendor $B$. Figure 2 visualizes the data collection scheme.

In total, we collected approximately 396 000 ads, 26 million longitudinal observations for 67 000 vendors with 4.7 million historical

reviews, and information on the more than 664 000 users that left that feedback, up to June 26, 2023. In Paxful, only 0.27% of all feedback is negative—comparable to 0.39% in the eBay US market[28].

We also collect data from LocalCoinSwap (LCS) from May 27, 2022, to June 26, 2023, using their public APIs. LCS API gives us all the posted ads, so that we have perfect coverage. In addition to ad data posted on the platform, LCS API allows us to query all the historical feedback data so we can get information on all the users who have given or received feedback at least once. In total, we collected over 52 000 ads, 14 000 users, and 146 000 feedback. Feedback is not binary, but on a 5-point scale; 1.7% are *below* 5.

Our user data corroborates anecdotal evidence that Paxful seems to attract a large proportion of customers from developing countries while LCS appears to attract more customers from western countries such as Europe and Australia (see Appendix A.1).

### 3.2 Ethics and legality of data collection

We collected data through publicly accessible APIs, abiding by both platforms' terms of service (ToS) as of the end of data collection. In particular, we did not scrape websites. Regrettably, the same ToS prevent us from redistributing the data we collected, but this paper should provide enough information about our collection methods for interested parties to attempt to reproduce our work. Our data do not contain personally identifiable information, so that our IRB(s) do(es) not consider this study human-subject research.

### 3.3 User suspension

To maintain safety, both platforms restrict or suspend users who violate their ToS. For Paxful, light violations (e.g., canceling a trade after its completion or using an outside app such as Telegram to conduct a trade without escrow) lead to restrictions being placed on the offending accounts. More serious transgressions lead to an immediate, permanent, and irreversible ban. Paxful lists four examples of such transgressions [34]: 1) using multiple accounts, 2) using fake identities, 3) accessing from OFAC-banned countries [32], or 4) using unauthorized gift cards, reversing payment, and defrauding users. LCS ToS [26] strictly prohibits "spoofing trades" – i.e., self-promoting attacks – to protect the credibility of the reputation system. From each user page, we identify whether a user is suspended based on API responses. (See additional details in Appendix A.2). Surprisingly to us, as many as 46% of all Paxful vendors in our corpus who posted ads are suspended (24 562 users out of 53 224 until March 1, 2023). Throughout this study, we consider suspended accounts as "riskier" accounts (i.e., which have committed one of the heavy violations described above). Since we rely on the platform to label the risky accounts we will use in our machine learning model (§5 and §6), we perform several additional validations of label quality. We check that 1) account suspension is at least partially handled by humans (and not through a purely automated process) and 2) most suspensions are permanent bans. Appendix A.3 contains details. To evaluate the level of current moderation effort, we estimate how long the platform takes to find malicious accounts after the creation of accounts; 18% are suspended within a week, 48% within a month, and 83% within a year. We also measure how long it takes to unban accounts that turn out to be benign; 32%

are unbanned within a week, and 68% within a month. Full details are in Appendix A.4.

## 4 EVALUATION ON EXISTING REPUTATION SYSTEM

In this section, we evaluate the current feedback-based reputation system used by both Paxful and LCS. We center our analysis on two guiding questions, derived from prior work: (1) *Does feedback convey enough information for customers to recognize risky vendors?* (2) *Is the reputation system trustworthy or is it susceptible to manipulation, such as self-promoting attacks?* [21, 28].

To address these questions, we conduct two empirical evaluations. First, we demonstrate that numeric (i.e., scores) and textual (i.e., reviews) feedback left about vendors is noisy and does not convey sufficient signal to properly assess vendor quality. Second, we identify the instances of self-promoting attacks and distill public signals that significantly differ between suspended and non-suspended accounts across both markets. We leverage these findings to inform the development of our prediction model in §5.

### 4.1 Feedback signals

We test whether the numeric and textual feedback conveys enough information for customers to discern potentially malicious accounts. Paxful shows the number of positive/negative feedback at the top of each user page. LCS displays the average feedback (on a five-point scale) on overall transactions for each user. To facilitate comparisons, we map these quantities to the [0, 1] range.

First, feedback is skewed towards perfect scores, which makes it harder for customers to distinguish between good and bad vendors. 96.43% of Paxful (resp. 95.48% of LCS) users have a feedback score greater than 0.95; and 90.89% of Paxful (resp. 84.67% of LCS) users have scores greater than 0.99. In other words, getting *one* negative feedback out of 20 transactions suffices to drop a vendor to the bottom 5%. This is not unique to cryptocurrency marketplaces: 96.5% of transactions were rated 5/5 in the Silk Road anonymous marketplace [11], and 90% of vendors have 98% feedback scores or higher in eBay [28]. To mitigate this skewness, Nosko and Tadelis [28] suggest EPP (Effective Positive Percentage), defined as the number of positive feedback divided by the number of total feedback. However, in Paxful, customers may conduct multiple transactions within a single listing, for which they can only leave one piece of feedback. Thus, EPP calculated on this platform is not comparable to that in previous literature.

We analyze the feedback text (i.e., reviews) next. We use the Google Translate API to translate into English approximately 8% of reviews in languages other than English. Through manual inspection, we identify six categories of negative feedback:

(1) **Scam accusations**: users sometimes explain fraud details, or simply call the vendor a scammer (e.g., *"He tried to rip me. Stay away from him," "Fake payment for payoneer invoice kindly don't trade this person. Return my amount 500 usd"*).

(2) **Complaints about speed**: being slow or unresponsive also leads to major complaints (e.g., *"Not fast," "I regret trade with him. 6hrs??"*).

(3) **Slander**: reviews that insult vendors without further details (*"Bad vendor," "Stupidity"*).

(4) **False negatives**: some reviews appear to be mistakenly registered as negative (e.g., *"Goodd," "Positive," "+++++++"*).

(5) **Quid-pro-quo**: ask/threaten trade partners to leave feedback in exchange for positive feedback (*"When you leave positive feedback I'll update mine," "selfish fello who doesn't leave a feedback after trade"*).

(6) **Unclear/other**.

To quantify the ratio between those categories, two of the authors independently manually labeled categories for 500 randomly selected negative reviews. For Coder 1 (Coder 2), 55.4% (46%) are scam accusations, 12.6% (10.4%) are about speed, 14.6% (22.4%) are slandering, 5.2% (5.0%) are apparent false negatives, 5.2% (3.2%) are quid-pro-quo, and 6.6% (13.0%) are others, respectively. The Cohen Kappa statistic [12], the agreement between two coders, is 0.706, which is considered "substantial agreement." Interestingly, even when manually annotating the data, extracting a clear signal from the text (or verify the credibility of reviews) is difficult, as observed by the disagreement between coders. In particular, coders had the most disagreements judging scam- and slander-related feedback. Furthermore, negative feedback tends to attract replies that rebut the reviews (e.g., *"As if it was very difficult to do what you did, you are very smart to make other people look bad"*). Indeed, 19.22% of negative reviews get a reply (compared to 0.71% in all comments), which implies that some negative reviews may be a form of retaliation or attempts to taint the reputation of competitors through a "badmouthing attack" [7] (e.g., *"stay away from him he will destroy your reputation he will mess you up after successful trade"*). As noted by the high skew of reputation scores, such retaliation attempts may be particularly effective against otherwise reputable vendors. In Appendix A.5, we automate the above classification using keyword search and apply it to the entire feedback corpus. Our observations suggest that obtaining a clear signal on the quality of a vendor either through their numeric reputation score and/or the reviews is difficult. This is further exacerbated by the fact that customers may often prefer to leave feedback outside of the platform due to retaliation fears [41] and employ external avenues, such as forums [14], or not even leave feedback at all due to the lack of economic incentives [37]. In our study, we observed users posting reviews on Reddit (e.g., /r/Paxful), Telegram (e.g., LCS Telegram channel), or even leaving negative reviews in app stores.

### 4.2 User collusion and automation

Our manual investigation reveals a set of accounts that exhibits the following traits. 1) More than hundreds of accounts are giving feedback together repeatedly. 2) Many *positive* feedback messages are submitted in a short range of time. 3) They reuse a similar set of simple feedback messages (e.g., *"Excellent trader very fast," "Good and quick"*) 4) They appear to arbitrarily pick rare payment methods, that are not currently in use in the account's origin country. 5) Many accounts share the exact same number of trade counts. Appendix A.6 describes the details, but this analysis is inspired by Fusaro et al. [18] that illustrates the unnatural distribution of trade volume as a sign of "wash trading" (creating fake trades by selling items to oneself to give the appearance of larger volumes) in centralized cryptocurrency exchanges. Based on those characteristics, we believe that these accounts engage in self-promoting attacks.

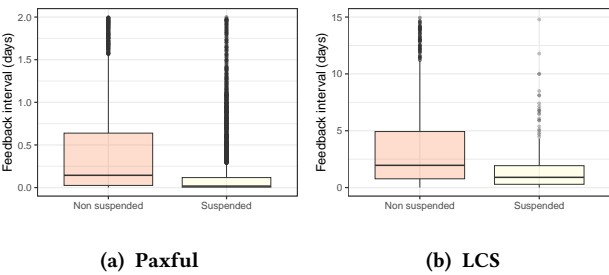

**(a) Paxful**                  **(b) LCS**

**Figure 3: Feedback interval (days) boxplots. We restrict the $y$-axis ranges for better visualization.**

Not only do these patterns suggest that existing reputation systems may be easily manipulated, they also hint at features that may be indicative of suspended accounts. We next look at features that suggest manipulative behaviors such as collusion and automation.
(1) **Interaction networks between accounts**: Consistent with other studies of online financial communication platforms [45], our data show that suspended accounts comparatively interact more frequently with other suspended accounts. Only 16.26% of the feedback from *non-suspended* accounts is directed towards suspended accounts, whereas 24.82% of the feedback from *suspended* accounts goes to other suspended accounts. Around 300 000 reviews – or 6.3% of all feedback observed – are generated between suspended accounts. This result motivates us to incorporate the information from neighboring users based on feedback interaction.
(2) **Feedback to transaction ratio**: suspended and non-suspended accounts also have unique differences in their feedback rates. Benign accounts often receive little feedback compared to the number of transactions they conduct—a predominant phenomenon across online marketplaces [41]. However, suspended accounts boast an unnaturally higher feedback rate. For instance, among accounts with 10 trades, 50.4% of suspended accounts received feedback for every transaction (i.e., they got 10 reviews); only 8.4% of non-suspended accounts were in that position. For 200 transactions, 6.7% of suspended accounts still get feedback for every transaction, whereas the number drops to 0.36% among non-suspended accounts. Thus, high feedback rates suggest possible user collusion.
(3) **Feedback interval**: Previous literature has shown that bots have a very different posting behavior from legitimate users [15], We investigate how frequently each user receives feedback. We define *feedback interval* as the median time between two consecutive reviews. We exclude accounts with less than 10 feedbacks due to noise. Figure 3 shows that suspended accounts received feedback far more frequently than non-suspended accounts. As an extreme example, one Paxful user received feedback every 4–5 seconds, which raises strong suspicions of automation.

## 5 PREDICTING ACCOUNT SUSPENSION ON PAXFUL

The above results answer the two questions posed earlier: existing reputation signals convey insufficient signals to determine the quality of accounts, and are easily manipulated by user collusion and automation. Furthermore, there are significant differences in features

besides feedback scores between suspended and non-suspended accounts. This suggests that other public signals, not captured by current reputation systems, can characterize problematic accounts. We next rely on these features to design a classifier, which can predict which vendors are suspended on Paxful. (We defer LCS to the next section.)

### 5.1 User features

We derive user features from four sources: user profiles/statistics, ads, and feedback. Feature selection is informed by our exploratory analyses in the previous section and by related work [15, 44, 45]. User profile and statistics include the number of users blocked by/blocking/trusted, registration time (Appendix A.4), registration country (given by IP address), number of trades, trade volume for each currency (BTC, USDT, ETH), number of trade partners, number of positive/negative feedback. We also keep track of users who access the platform from countries different from where they initially registered (Appendix A.1).

For listings, we aggregate all the collected ads at the user level (e.g., posting 60% of ads in USD makes "ratio of USD in ads" variable equal to 0.6). An important feature derived from user ads is the price premium, defined as the difference between the advertised price and the market price, i.e.,

$$\text{Price premium} = \frac{\text{Proposed price} - \text{Market price}}{\text{Market price}}.$$

Prior work on Craigslist has found that scammers often set unreasonably low-price premiums [29, 30, 44], which motivates using it as a feature. Other ad data include timezones (based on the city listed in the ad), payment method (e.g., bank transfer, PayPal, Amazon gift cards), types of fiat currencies (e.g., USD, EUR, KES), cryptocurrencies (e.g., BTC, USDT), any customer verification requirements, and whether users are marked as "verified" by the platform [31]. We further compute feedback interval (§4.2), and incorporate the negative feedback content identified by keyword search (see §A.5).

Finally, we build a feedback graph (i.e., each node is a user and a directed edge $A \rightarrow B$ is feedback from user $A$ to user $B$) to include neighbor information. Since feedback is not mandatory, the feedback graph is a strict subset of the entire trade graph. From this graph, we derive network metrics such as ego density and some centrality measures to incorporate how they interact with others, and how influential they are. Importantly, users are allowed to change their username *only once* on Paxful. We keep track of those changes and reflect them when we aggregate all the features. We normalize all features (mean 0, std. dev. 1) to stabilize model training, except for binary variables and features already in [0, 1].

### 5.2 Machine learning models

Using the labels described in §3.3, we build a machine learning model to classify suspended accounts between suspended (24 562) and not-suspended (28 662). Our model construction is inspired by prior bot detection work (e.g., Davis et al. [15]). We implement seven machine learning models: Logistic Regression, K-Nearest Neighbors, Decision Tree, Random Forest, XGBoost [10], LightGBM [23], and Neural Network, using Python *scikit-learn* to compare their performance. We use grid search with 5-time cross-validation to tune

**Table 1: Prediction results of the seven models with confidence interval (2.5%, 97.5%)**

|  | Accuracy | Precision | Recall | F1 | AUC |
|---|---|---|---|---|---|
| Logistic Regression | 0.769 (0.758, 0.781) | 0.768 (0.757, 0.78) | 0.767 (0.755, 0.778) | 0.767 (0.756, 0.779) | 0.849 (0.838, 0.859) |
| K-Nearest Neighbors | 0.775 (0.764, 0.786) | 0.779 (0.768, 0.79) | 0.779 (0.768, 0.79) | 0.775 (0.764, 0.786) | 0.86 (0.85, 0.87) |
| Decision Tree | 0.818 (0.808, 0.829) | 0.817 (0.807, 0.828) | 0.818 (0.808, 0.828) | 0.818 (0.807, 0.828) | 0.879 (0.87, 0.889) |
| **Random Forest** | 0.856 (0.847, 0.866) | 0.859 (0.849, 0.868) | 0.853 (0.844, 0.863) | 0.855 (0.845, 0.865) | 0.931 (0.924, 0.937) |
| **XGBoost** | 0.862 (0.853, 0.871) | 0.862 (0.853, 0.871) | 0.861 (0.851, 0.87) | 0.862 (0.852, 0.87) | 0.935 (0.929, 0.941) |
| **LightGBM** | 0.861 (0.852, 0.87) | 0.862 (0.852, 0.871) | 0.859 (0.85, 0.868) | 0.86 (0.851, 0.869) | 0.932 (0.925, 0.938) |
| Neural Network | 0.825 (0.815, 0.835) | 0.824 (0.814, 0.834) | 0.826 (0.816, 0.836) | 0.825 (0.814, 0.835) | 0.903 (0.895, 0.911) |

model hyper-parameters/architectures (e.g., the level of regularization, the depth/number of trees, and the number/dimensions of neural network layers). We divide the entire dataset into 80% training/validation set and 20% test set, and use the test set to conduct out-of-sample prediction and compare performance. Table 1 summarizes the results of each model for accuracy, precision (macro), recall (macro), F1-score (macro), AUC (area under the curve), with a threshold of 0.5. Ensemble-based tree algorithms (Random Forest, XGBoost, and LightGBM) outperform other methods, achieving 0.86 F1 and 0.93 AUC. To draw statistical differences between models, we randomly pick 50% of test sets, bootstrap for 10 000 times and derive the (2.5%–97.5%) confidence intervals shown in parentheses in Table 1. For example, for Random Forest, the F1-score falls in the 0.842–0.862 range for 95% of bootstrapping. Based on it, we conclude that the three ensemble tree-based algorithms (Random Forest, XGBoots, LightGMB) perform equally well while significantly outweighing the others.

To delve into how our model identifies risky accounts, Table 2 (Paxful: first column) highlights the top-10 most important features for tree-based ensemble models. This is calculated based on how good the split is ("gain") when using each feature. The most important source of information is the number of accounts the user is blocking, which is a good proxy for how adversarial the account is. Models also seem to rely on various sources of data including user profiles (registration time), trade statistics (number of positive feedback, number of trade partners), ads information (price premium, currency), and network metrics from feedback graphs (ego density). Some of the features, e.g., pricing strategies [5] and ego density [45] were found to be characteristic of suspicious accounts in previous literature. A number of trades, positive feedback, feedback interval (ranked in the top-15 features), and network metrics are frequently associated with user collusion and feedback automation (§4.2). "Verified" user badges, on the other hand, have little impact on our model's decision-making (not in the top-50 features); this echoes other studies [46, 49]. In short, integrating multiple sources of public information, rather than merely assessing reputation through feedback scores and/or badges, appears desirable.

### 5.3 Evasive measures
Our machine learning model presents a few limitations that malicious participants could potentially exploit.

**Table 2: Top 10 most important features for Paxful (§5) and LCS (§6) categorized by data source. Number in parentheses is the feature rank.**

|  | Paxful | LCS |
|---|---|---|
| User profile | Number of user blocking (1) Registration time (2) Number of users trusted by (9) | Registration time (1) Number of users trusted by (8) Number of users blocked by (9) |
| Trade statistics | Number of trades (3) Ratio of positive feedback (5) Number of trade partners (6) | Number of trade partners (3) Number of trades (4) Average response time (7) |
| Ads | Price premium (4) Ratio of USD (10) |  |
| Feedback | Ego density (7) Total degree (8) | Eigenvector centrality (2) Total degree (5) Ego density (6) Feedback receiving interval (9) |

First, assuming that an attacker knows the detailed implementation of our machine learning model, they can control some parameters to avoid detection. For example, they can avoid using certain types of payments (e.g., PayPal, M-Pesa), or types of currencies or coins (e.g., USD, KES, BTC). An attacker could use a VPN to obfuscate their location (see Appendix Figure 6) if they are aware that the model tends to pick more users from a certain country. Our model also fails to capture users who rely on new or unpopular types of payment, currencies, or locations. On the other hand, changing those would make it much harder for an attacker to attract legitimate customers. In other words, evasion, while possible, could come at a potentially hefty price to the attacker.

Second, some features (e.g., the number of users being blocked) may slowly evolve, and a malicious participant could exploit the time lag before they get flagged. However, this latency also applies to feedback-based reputation (trades need to be completed for feedback to appear), and our model is less susceptible to it since it combines multiple features.

Third, the model is vulnerable to whitewashing attacks [21]. If a scammer creates a new account to purge their entire history, the model will fail to identify them, at least initially. However, this too comes at a cost: reputation needs to be rebuilt from scratch.

## 6 GENERALIZING THE MODEL ACROSS MARKETS
To test the generalizability and transferability of our models across platforms, we repeat the previous experiment beyond Paxful, varying features, training sets (Paxful vs. LCS), and prediction targets (Paxful vs. LCS as well) to generate six different models (Model 1–6) for testing. For the sake of simplicity, we limit our use to Random Forest (one of the best-performing models in Table 1) in this section. Table 3 summarizes our results for these six models as described below. Our baseline, Model 1, is the model described in the previous section (Paxful).

From historical feedback LCS data, we extract 11 657 accounts. For those, we check the user page status and found 1 547 (13.27%) suspended accounts. In LCS, account information becomes unavailable after users get suspended. As a result, we can only collect profile data for 167 suspended accounts. To account for this data

**Table 3: Performance results for two markets (Susp. = num. of suspended accounts, Acc. = Accuracy)**

| Model | Data | | | | | Performance | | | | |
|---|---|---|---|---|---|---|---|---|---|---|
| | Training | Features | Prediction | Test size | Susp. | Acc. | Precision | Recall | F1 | AUC |
| 1 | Paxful | All | Paxful | 10645 | 4935 | 0.858 | 0.860 | 0.855 | 0.857 | 0.931 |
| 2 | LCS | All | LCS | 260 | 38 | 0.869 | 0.773 | 0.596 | 0.624 | 0.684 |
| 3 | Paxful | Common | Paxful | 10645 | 4935 | 0.723 | 0.723 | 0.719 | 0.719 | 0.791 |
| 4 | LCS | Common | LCS | 260 | 38 | 0.858 | 0.712 | 0.557 | 0.567 | 0.638 |
| 5 | Paxful | Common | LCS | 1300 | 169 | 0.840 | 0.600 | 0.566 | 0.576 | 0.659 |
| 6 | Paxful | Common | LCS | 367 | 169 | 0.594 | 0.647 | 0.565 | 0.510 | 0.632 |

loss, we downsample the non-suspended accounts to keep the suspended and non-suspended ratio identical (13:87) to the original population. We repeat the same procedure described in §5 for LCS, and use the data prior to March 1st, 2023 to temporally align with our Paxful experiment. Since available user information differs from what Paxful provides, we use different features in LCS such as the average response time, and primary currency/language. Model 2 is identical to Model 1, but independently trained on LCS data (and predicting on LCS). Model 2 does not achieve the same performance level as Paxful (Model 1). This is probably due to the smaller number of data samples, fewer features, and imbalanced label distributions. To test model generalizability, we then only use features common to both platforms. These include feedback interval, trade counts, negative feedback ratio, the number of trade partners, and network metrics such as ego density on the feedback network. We do not normalize features. First, we re-train the model with those common features on Paxful data (Model 3) and on LCS data (Model 4). Model 3 does not quite manage to match the performance of Model 1; on the other hand Model 4's performance is roughly the same as Model 2's, despite the smaller number of features. This indicates some features only (publicly) available in Paxful, such as the number of users being blocked by the user, are crucial to performance.

Finally, to test model transferability, we first train the model using Paxful data, freeze the model weight and make predictions for all the users on LCS (i.e., using both train and test data on LCS) (Model 5). Since Paxful has a larger number of samples than LCS, we should observe a performance increase in LCS if we can successfully transfer some knowledge from Paxful. Unfortunately, the performance does not significantly improve from simply training on LCS independently, which means the model does not appear to be directly transferable from Paxful to LCS within our dataset. To explain why, we consider three factors. First, the proportion of suspended accounts is 46% in Paxful but 13% in LCS, so the model might have been confused. To test this conjecture, we downsample the non-suspended accounts to keep the ratio identical to Paxful's (Model 6), but do not observe any increase in performance. Second, the user base is markedly different (see §2.1, §3.1, and §A.1). Third, both platforms operate at different scales. Paxful has *at least* 4.7 million reviews whereas LCS has only about 146 000 reviews; however, normalization of the features does not alleviate this issue. On a more positive note, we find, in Table 2, that some features, indicative of risky accounts, are important to both platforms, such as network metrics (ego density) and trade statistics (number of trade partners, trade counts, feedback interval).

## 7 ONLINE EVALUATION

The main use of a model such as we propose is as an early warning for platform administrators and moderators that certain users might be suspicious, so that they can focus limited resources on truly problematic cases. In this section, we develop a framework for monitoring users, using 30 days of user profile data.

### 7.1 Experimental setting

We perform an online evaluation over March 1–March 30, 2003. From the active users as of March 1, we create three sets of 500 users each: 1) the "riskiest" users according to our machine-learning model prediction "ML," 2) users with the lowest reputation, "REPUTATION (REP)," and 3) randomly chosen users, "RANDOM (RND)." For ML, we define the "riskiest" users as the set of users that have not been suspended yet, but that our machine learning model predicts will be suspended with the highest probability. We train our model using data until February 28th, 2023. For REP, we choose the users with the highest ratio of negative feedback and at least 10 reviews. We check each user at least once every day for suspension and trade count. There is some overlap in users between each group, so, to keep independence assumptions, we exclude these common users when performing statistical tests. We set the $p$-value statistical threshold to 5%, and apply Bonferroni correction to account for multiple hypothesis testing.

### 7.2 Results and Implications

Table 4 (left) summarizes overall results, showing that around 20% (95 users) of users in ML get suspended within 30 days. A pairwise $\chi^2$ test confirms that ML predicts significantly more suspensions than REP or RND, while there is no statistical difference between REP and RND – see the first column in Table 4 (right). In other words, a reputation system solely based on feedback does little better than random; instead, a larger set of features and a classifier similar to ours helps predict suspensions far more accurately.

Next, we discuss the timing of user suspension. Figure 4 delineates the survival curves for each group, that is the number of users initially active on March 1, 2023, that remain active on the platform over the 30 days of evaluation. The survival curve for ML decays much faster than the other two groups. In other words, "risky" users according to the ML prediction are much more likely to be suspended soon. We confirm the statistical difference between ML and the other two models using a log-rank test (non-parametric)— see the second column in Table 4 (right). The result suggests that our model is able to identify risky accounts that have not yet been flagged by the platform (i.e., false negatives) earlier.

Besides suspensions, we measure the number of trade completions, which is a good proxy of how successful and active users are. We conjecture that there is a negative impact from a low reputation score on the amount of trade. To account for the fact that some users get suspended in the middle of the observation period, we divide the total number of completed transactions during this experiment by the number of active days over the month. Using a $t$-test, we find a significant difference between RND and the other two groups, indicating that risky users from the ML model and the low reputation group complete fewer transactions – see the third column in Table 4 (right). In other words, Although our machine

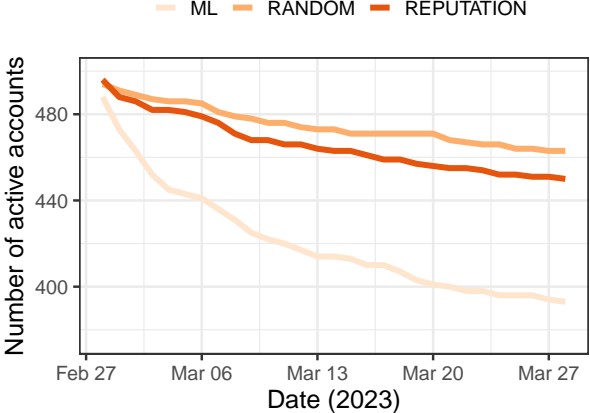

**Figure 4: Survival curve (i.e., the number of active users) for each group from March 1st. $y$-axis starts at 375. Curves do not start at 500 because we exclude users who changed usernames.**

**Table 4: Results of online evaluation for three groups (left), test-statistics ($p$-value) for each pair (right): (Susp.= num. of suspended accounts, Trade = num. of trades completed per day, Survival = survival curves)**

| | Final outcome | | | Statistical test | | |
|---|---|---|---|---|---|---|
| | Susp. | Trade | | Susp. | Survival | Trade |
| ML | 95 | 16.13 | ML-REP | 20.14 (0.000) | 21.99 (0.000) | 2.04 (0.041) |
| REP | 46 | 9.15 | ML-RND | 40.98 (0.000) | 43.44 (0.000) | -2.76 (0.006) |
| RND | 30 | 38.83 | REP-RND | 3.21 (0.073) | 3.63 (0.057) | -4.29 (0.000) |

learning model is optimized to predict account suspension, it can, to some extent, identify unsuccessful vendors. Unsurprisingly, users with poor feedback tend to be less successful on the market too.

To validate the robustness of our method, we perform three additional experiments to check 1) how much variance exists when randomly picking users (RND group), 2) whether the result changes using a different timeframe, and 3) the optimal number of users to pick (i.e., not fixing it to 500 users). All the experiments confirm the superiority of our ML method. We refer to Appendix A.7 for detailed implementations.

## 8 RELATED WORK

This section relates our work with previous efforts on 1) cryptocurrency P2P exchanges and 2) online misbehavior in other platforms, and highlights the novelty of our research.

The cryptocurrency P2P marketplace landscape largely remains understudied. In LocalBitcoin, Von Luckner et al. [48] identified many transactions as remittances from the US to developing countries. Andreianova et al.'s survey [6] further clarified that many users from Latin America use P2P platforms for remittances, whereas

users in Africa use the platform for trading/profit generation. Van de Laarschot and van Wegberg [47] connect online anonymous market vendors to major P2P cryptocurrency exchanges. However, despite their relevance, no prior work has evaluated the online safety of cryptocurrency P2P markets.

On the other hand, some empirical studies look at scams in other marketplaces such as Craigslist [5, 19, 29, 30, 44]. A common detection approach is to use the platform-provided labels [19, 30] and complement them by unrealistically low price premiums [5, 44], or directly interacting with suspicious accounts [29]. We choose the first approach to discover suspicious accounts and perform some validations to confirm label quality (i.e., a low number of false positives), and additionally extend our analysis to multiple markets for generalization. We further develop a platform monitoring scheme to prove the practicability of our method as well.

Another related line of work revolves around user misbehavior on social media platforms, particularly social bot detection [15, 16, 20, 52] using machine learning on large-scale data [16]. In particular, our work adopts a similar methodology to Davis et al.'s work [15] on feature selection and algorithms. Others have studied account suspension [4, 43], and shown that fake/suspended accounts form closely knit communities [9, 24, 45, 51], which our study confirms.

## 9 CONCLUSION AND DISCUSSION

This paper is the first to investigate online misbehavior in cryptocurrency P2P marketplaces. We outline the limitations of solely relying on feedback-based reputations and attempt to build a better system for uncovering risky vendors. Using only publicly available data, our model achieves 0.93 AUC in identifying account suspension in Paxful, one of the most active cryptocurrency P2P marketplaces. We expect the performance would increase with access to private information such as IP addresses, especially on a smaller platform like LCS. We could not replicate our experiments on other platforms such as Binance P2P, which do not provide indicators of account suspension. In practice, with access to internal back-end data, any marketplace can follow the same procedure and incorporate features we identified as important. We further provide a framework to improve platform moderation. Instead of directly banning the accounts the model identifies, we suggest selecting the set of accounts with the highest likelihood of suspension and prioritizing them for monitoring.

Our results also benefit users. Our study shows users should review various types of features besides feedback, such as price premiums, and who is giving feedback. Note that our model does not protect users after they initiate transactions (i.e., only help identify risky vendors as a precautionary measure). After starting a trade, platforms recommend users verify payments in addition to the receipt sent by counterparties, take screenshots frequently to gather evidence, and avoid outside channels to communicate [8].

More generally, our work helps broader research on other online marketplaces that remain understudied (e.g., gift cards, NFT, online loans). Those platforms rely on reputation systems and face issues similar to what we observe. Another research area lies in reputation system design (i.e., how to convey the risks associated with vendors) since the way the platform aggregates/presents reputation scores significantly affects user behavior [37].

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

## A  APPENDIX

### A.1  Geographical considerations

User origin is another feature we consider for our experiments. For each platform, we aggregate the number of users by country of origin. In Paxful, the country seems to be determined by the IP address used when registering an account. In LCS, users self-disclose their local currency, so we employ this as a proxy for their location. The default is set as USD. Figure 5 shows the number of users for each country: Paxful on the left and LCS on the right. The customer base seems to be significantly different between both platforms. Paxful features many users from Africa, such as Nigeria (NG), Kenya (KE), and Ghana (GH), while LCS attracts more users from Australia (AUD) and Europe (EUR).

At the time of data collection, the Paxful API returns both the country of registration and the country from which the user last accessed the platform, based on the user's IP address. Our long-term observations reveal that some vendors appear to log in from countries different from their country of registration. Figure 6 is a heatmap that evidences these changes. The $y$-axis is the country of registration and the $x$-axis is the country of access (any point in our observation). For better readability, only include pairs of countries with more than 30 distinct users, and normalize by the $x$-axis (number of accesses). We observe that many users route through the US, Kenya, and Nigeria. Given that these users registered in a different country, we hypothesize some of their traffic is over VPNs (or Tor) to obfuscate its true origin. Figure 7 shows the ratio of users, per country, who access the site from a different country at least once. We only include countries that have more than 500 vendors. For example, more than 99% of vendors who registered in China later used IP addresses from a different country. Users in cryptocurrency-regulated countries such as China (CN), Bangladesh (BD), Indonesia (ID), Pakistan (PK), Vietnam (VN), and Cameroon (CM) [42] appear to connect to the site from alternate locations often. These users are incentivized to obfuscate their location, but do not seem to maintain good operational security in the long run. Another possible motivation is to circumvent restrictions that Paxful has for certain

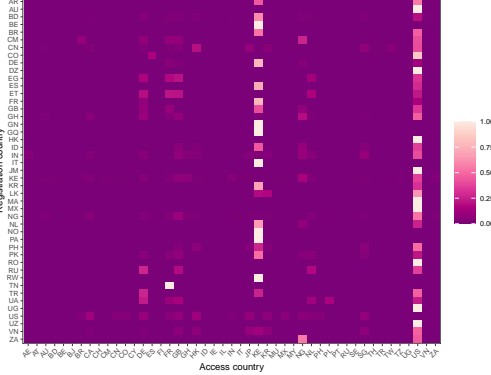

**Figure 6: Heatmap of country changes between registration and subsequent accesses (normalized by $x$-axis).**

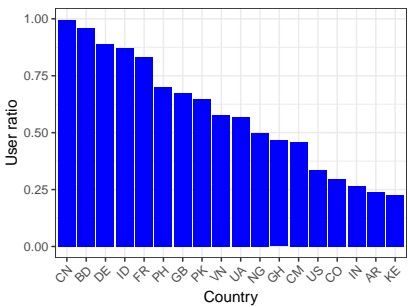

**Figure 7: Ratio of users for which the reported country of access is different at least once from the registration country (for countries with more than 500 users).**

payment methods in some jurisdictions. For example, as reported in the Paxful subreddit [35], Zelle is prohibited in Cameroon, China, Ghana, India, and Nigeria at the time of writing. However, some of these users might also simply be traveling.

### A.2  Identifying suspension

We rely on values returned by the API(s) to distinguish between regular users, suspended users, and users who have changed their usernames. In Paxful, upon suspension, a user is marked as "not active" on the web page and the API call for their profile returns a JSON field "is_active" as False. According to a Paxful moderator on Reddit [36], this indicates either an account ban (non-reversible) or an account lock (reversible). In terms of account deletion, Paxful API does not appear to change.

Unlike Paxful, LCS does not explicitly mark accounts as suspended, a user page is taken down when the user changes to a different username, or when they get suspended by the platform. The page says "not found" when the username has changed, but redirects to the ads page if the user was suspended. Likewise, the API responds differently. We attempted to collect account suspension data from other platforms such as Binance P2P, one of the largest players in this space, but could not identify the signs of account suspension on those. Indeed, a Binance P2P user page does

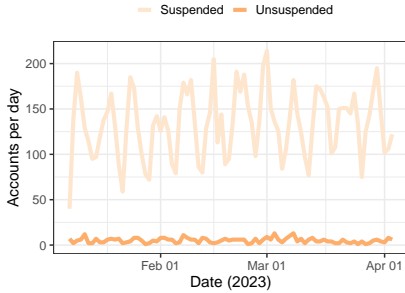

**Figure 8: Account (un)suspensions per day on Paxful, 01/08/23–03/31/23.**

not seem to change even if the user deletes the account. We use account suspension as the main label (and prediction target) of our machine learning model in §5 and §6.

### A.3 Suspension label validation (Paxful)

We performed several validations to ensure the quality of the suspension labels obtained from Paxful. Figure 8 shows the number of suspensions and unsuspensions for each day between January 8th, 2023 and March 31st, 2023. We only include users for whom we can confidently determine the time of the suspension. More precisely, we pinpoint the time of suspension if the user ban status changed from false to true based on two consecutive observations within one day (86 400 seconds). We can confirm some weekly seasonality—there is a decrease in the number of suspensions on weekends—suggesting that platform moderation is not purely automated, and instead relies on human input to some extent. Second, the label seems to imply permanent suspension for most users. Longitudinal observations confirm that only a small portion of those are unsuspended (lower curve).

### A.4 Platform moderation evaluation

To investigate the level of platform moderation, we evaluate how long the platform takes to find malicious accounts and how long it takes to lift suspensions on accounts that turned out to be benign.

We first calculate the number of active days for suspended accounts. Paxful API returns a rough estimate of registration time (e.g., "3 hours before" or "1 month before"). We monitor changes in that response across queries (e.g., "4 days before" to "5 days before"), and estimate the registration timestamp based on multiple data points. Figure 9a shows the Cumulative Distribution Function (CDF) for the ages of suspended accounts. 18% of suspended accounts are suspended within one week of registration, 48% within a month, and 83% within a year. The small spikes around 365 and 730 days are an artifact of the coarseness of our estimated registration time, which becomes less accurate for old accounts (e.g., the API returns "1 year ago" to "2 years ago").

We next derive the number of days the platform takes to lift suspensions on accounts that turned out to be benign. Minimizing the length of an erroneous suspension is critical to building trust with customers. To measure this, we select users who have been suspended once but were unsuspended later. We calculate the time

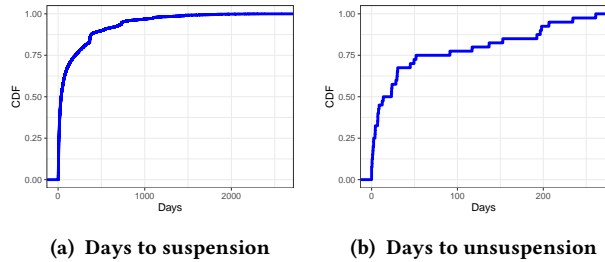

| (a) Days to suspension | (b) Days to unsuspension |

**Figure 9: CDFs of the number of days to suspension and to release.**

span between the observation when they first get suspended and one observation before the status changed to "active" (i.e., a lower bound). Figure 9b shows the CDF of the length of time before the status of a suspended account is restored. We only include accounts in which we can confidently identify the timing of the unsuspension ($n = 41$). Within a week of (erroneous) suspension, 32% of these accounts see their bans lifted; 68% are unsuspended within a month. We do not include accounts that have not been released at the end of our observation period.

### A.5 Feedback keyword searches

To find scam-related feedback at scale and spot risky users from feedback comments, we come up with a list of keywords: "scam," "rip," "liar," "conman," "thief," "thieves," "crime," "criminal," "fraud," "steal," "stole," "cheat," "fake," "ghosted," "swindle," "chargeback," "reverse," "coin locker" and perform a keyword search (perfect match) to discover scam-related feedback for all the negative reviews we collected. We used the same procedure for slow vendors: "slow," "sluggish," "not fast," "not responsive," and "delay" as well. We try to avoid false positives, i.e., to avoid flagging non-scam-related reviews as scams. To test the efficiency of our keyword-based approach, we run the keyword search on 500 reviews annotated by our first coder from §4.1 as validation. 41% of these reviews are captured as scam-related feedback with zero false negatives. The coder annotated 55.4% of these as "scam," meaning our automation failed to detect 14.4% of scam-related feedback. We thus regard the result of the keyword search as a lower bound.

Among all negative reviews, our automated classification flags 40% as scam-related feedback, and 9.45% of transactions were speed-related. By aggregating at a vendor level, 2 493 users have at least one scam-related feedback, and only 642 users (around 2.6% of total suspended accounts) received multiple scam-related feedback. Considering that a total of 24 562 (46%) vendors are suspended, solely looking at feedback data fails to spot many risky accounts. Nevertheless, we incorporate the number of scam/speed-related keywords as one of the features in our ML model – but realize it is not sufficient on its own.

### A.6 Complementary evidence of self-promoting

This section provides complementary discussion about users that appear to engage in self-promoting attacks described in §4.2.

First, those users pick rare payment methods, that do not appear to be used in their country of registration. Figure 10a shows

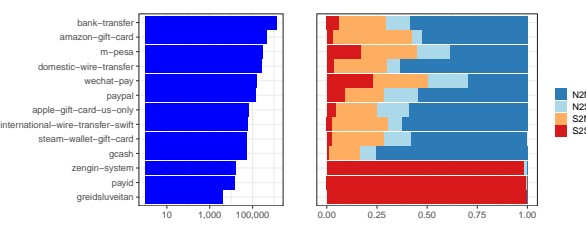

**(a) Number of feedback**     **(b) Frequency of interactions**

**Figure 10: Top 10 payment methods + 3 selected payments. N: non-suspended accounts, S: suspended accounts (e.g., N2S: feedback from non-suspended to suspended accounts)**

the number of reviews for 13 payment methods: the top 10 payment methods (bank transfers, Amazon Gift card, M-Pesa, etc.), and three payment systems we choose to investigate: Zengin (Japan), PayID (Australia), and Greidsluveitan (Iceland). Figure 10b further shows the split among these reviews for four interaction types: N2N (non-suspended accounts giving feedback to non-suspended accounts), N2S (non-suspended to suspended accounts), S2N (suspended to non-suspended) and S2S (suspended to suspended). The three payment systems at the bottom are dominated by suspended-to-suspended transactions. Looking at these three payment systems, 8 563 unique users give feedback, and only 391 users receive feedback. Interestingly, all the users giving feedback are from Vietnam – and not Japan, Australia, or Iceland where those three payment methods are reportedly used.

We further confirm that a subset of more than 100 of these users giving feedback send feedback together repeatedly. Those users appear to have been solely created for the purpose of self-promoting attacks, that is, they appear to be Sybils tasked with boosting the reputation of the feedback receivers. For example, one user received feedback from those 103 users through the Zengin payment system. All feedback was sent within 1 400 seconds and all reviews were positive. Several variations of the same comments appear to have been re-used (e.g., *"Excellent trader very fast.," "Good and quick," "Welcome to trade with me again," " He is a reliable trader."*).

In addition, those accounts exhibit unnatural trade distributions. The trade count of the users giving feedback is oddly distributed. For example, among all users that rely on the Zengin payment system, five accounts have engaged in three trades or less, 300 users have exactly four trades, but only two users engaged in five trades. This strongly suggests the presence of Sybils and automation. Similar findings apply to the other two payment methods. Most users receiving feedback have between 200–250 trades, which is markedly different from the overall distribution of trade counts. Based on all of the above, we believe these accounts are most likely engaged in coordinated self-promoting attacks.

### A.7 Robustness tests for online evaluation

To confirm the robustness of our evaluation, we conduct three additional experiments.

In §7, we randomly pick 500 accounts out of over 28 000 active accounts for the RND method, but we do not know how the results vary depending on the users we pick. To address this, we randomly

**Table 5: Results of online evaluation and statistical tests. Notation follows Table 4.**

| | | | | Statistical test | | |
|---|---|---|---|---|---|---|
| | **Final outcome** | | | Susp. | Trade | Survival |
| | Susp. | Trade | | | | |
| ML | 104 | 10.78 | ML/RP | 16.46 (0.000) | 3.33 (0.001) | 17.86 (0.000) |
| RP | 49 | 6.09 | ML/RN | 54.43 (0.000) | -2.29 (0.022) | 57.60 (0.000) |
| RN | 26 | 159.99 | RP/RN | 11.84 (0.001) | -3.25 (0.001) | 13.02 (0.000) |

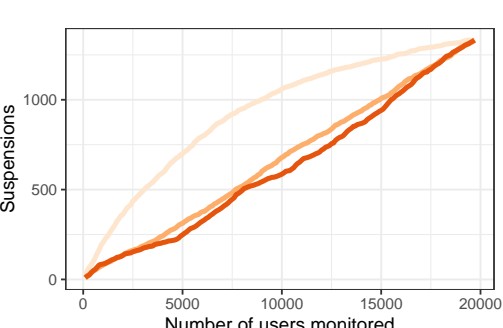

**Figure 11: Number of users monitored ($x$-axis) and number of suspensions ($y$-axis).**

draw 500 accounts, count the number of suspensions in a month, and repeat the process 10 000 times to check for any deviations in the results. More than 95% of the time, 18–39 out of 500 RANDOM accounts end up being suspended; in other words, our results in Section 7 about the significantly superior performance of ML/REP holds across many RANDOM samples.

Second, we perform the same online evaluation on a different time period: 30 days from February 1st, 2023. Our ML model is trained using only data before January 31st, 2023. Table 5 summarizes the outcome, which is consistent with the result presented in the main body —for a time interval starting on March 1st, 2023.

Third, in our online evaluation in §7, each group is the 500 riskiest/low reputable/random users. Here, we calibrate the number from 100 to 28 000 users. In other words, we monitor $x$ riskiest/low-reputation accounts and vary $x$ instead of fixing $x = 500$, and quantify the impact of $x$ on the number of suspensions. Figure 11 shows the number of total suspended accounts in a month ($y$-axis) based on the number of users monitored ($x$-axis). Obviously, if each method selects all active 28 000 vendors, all (ML, REP, and RND) methods have the exact same number of suspensions (i.e., the right top of the figure). However, the figure clearly illustrates that ML outweighs other methods regardless of the number of users monitored. It works best until around 10 000 users. Depending on the number of moderators the platform employs, they can adjust the number of users being monitored; the advantage of using our ML method marginally decreases when a large number of users are monitored.