# OpenReview forum: "Identifying Risky Vendors in Cryptocurrency P2P Marketplaces"
_ACM.org/TheWebConf/2024/Conference — TheWebConf24_

### Official Review · Reviewer_NHY7 · 2023-11-14

**Novelty:** 5
**Technical Quality:** 6

**Review:**

The paper studies datasets from two online cryptocurrency marketplaces, Paxful and LocalCoinSwap, studying reputation mechanisms (vendor feedback) and account suspensions. Since transactions may involve non-cryptocurrency payments (giftcards, etc.) there is a risk of fraud. The dataset from Paxful is particularly comprehensive, spanning more than 1 year, collected through their public API. Several questions are investigated, including:
- How well can machine learning methods predict account suspensions? Here, the best method achieves 93% AOC which seems quite good
- Is there evidence of reputation manipulation such as self-promotion? Several signs of such manipulation is identified

The paper collects what seems to be a unique dataset that may be of interest beyond the particular setting of cryptocurrency marketplaces. The studies conducted do not seem particularly deep, but are interesting and could be a starting point for further work, especially if the datasets are eventually released.

Since cryptocurrency marketplaces can be used for criminal activities it might be good to do an ethics review of the paper, though I do not understand the space well enough to say if there may be issues.

**Questions:**

- Though the ToS does not allow you to release the datasets, did you try to contact the platforms to get special permission? These platforms may have an interest in research conducted on their data, and it would improve the impact of your work if others could follow up on the same datasets.

**Ethics Review Description:**

It is unclear to me to what extent platforms such as Paxful and LocalCoinSwap are used for criminal purposes. IF this is a significant activity, it should be considered whether this study might indirectly support criminal activities.

**Ethics Review Flag:**

Yes

**Reviewer Confidence:**

2: The reviewer is willing to defend the evaluation, but it is likely that the reviewer did not understand parts of the paper

**Scope:**

4: The work is relevant to the Web and to the track, and is of broad interest to the community

---

### Official Review · Reviewer_878T · 2023-11-24

**Novelty:** 5
**Technical Quality:** 5

**Review:**

The paper analyses online safety within cryptocurrency peer-to-peer (P2P) marketplaces, specifically focusing on the identification of risky vendors. The authors gathered data from two cryptocurrency marketplaces over one year, encompassing millions of historical feedback entries. Subsequently, they developed a machine learning (ML) model to predict account suspensions, showing promising results.

I found the paper well-written and easy to read. It tackles an interesting problem, with several strengths:
1. The dataset is robust and extensive. The use of publicly available APIs likely ensures data completeness.
2. The literature review is well-constructed, providing good coverage of P2P crypto marketplaces and reputation.
3. The findings are interesting, including the potential manipulation of the current reputation system, and internal back-end data can help better reputation evaluation.
4. The coding for manual inspection of negative feedback categories is appropriate, supported by a considerably good Kappa score.
5. The comparison of different ML models is commendable. The authors also did an online evaluation by monitoring the vendors for 1 month, with appropriate statistical tests in Section 7.

**Questions:**

1. Given the extensive nature of the dataset, can the authors explore how vendors overcome the cold-start problem to establish their reputation in an untrusted setting?
2. Statistical tests should also be used in Section 4 to confirm the significant difference between "non-suspended" and "suspended" accounts for feedback rate and feedback interval within and across different marketplaces.
3. The 30-day evaluation is appreciated, however, “20% (95 users) of users in ML get suspended within 30 days”, does this mean the rest 80% did not follow their predictions?
4. When testing suspended users, they remove overlapping users between groups to satisfy the independence assumption, but could they use a different test without that assumption?
5. Please do not claim anything "first". Due to the breadth of existing literature, parallel work may have been conducted.

**Reviewer Confidence:**

3: The reviewer is confident but not certain that the evaluation is correct

**Scope:**

4: The work is relevant to the Web and to the track, and is of broad interest to the community

---

### Official Review · Reviewer_jagy · 2023-11-28

**Novelty:** 5
**Technical Quality:** 5

**Review:**

This work explores the risk involved in peer-to-peer (P2P) cryptocurrency exchanges, focusing on Paxful and LocalCoinSwap. These exchanges prioritize privacy but face higher risks of scams. The study finds that current feedback systems inadequately assess risky vendors and are susceptible to attacks and manipulation such as self-promoting, whitewashing, retaliation, and bad-mouthing. To address this, the researchers employ machine learning models, achieving high predictive accuracy for account suspension on Paxful. The models show limited transferability across platforms, emphasizing the importance of platform-specific features. Evaluations show that the models outperform traditional feedback-based systems at predicting which users will be suspended in the future.

Strengths:

S1. First work to study the safety of P2P cryptocurrency market place.

S2. Proposed solution improves over existing feedback system

Weaknesses:

W1. Lack of generality and transferability in the proposed solution.

**Questions:**

Are there any P2P marketplace that requires proof of transaction for submitting feedback or some other strict moderations for submitting feedback? How does the proposed solution compare with such strict feedback systems?

**Reviewer Confidence:**

3: The reviewer is confident but not certain that the evaluation is correct

**Scope:**

4: The work is relevant to the Web and to the track, and is of broad interest to the community

---

### Official Review · Reviewer_3kF6 · 2023-11-28

**Novelty:** 4
**Technical Quality:** 3

**Review:**

This paper presents a study to evaluate online safety and trust in cryptocurrency P2P Marketplaces. Specifically, the authors collected data for a year to build a dataset and based on their built dataset, the authors empirically show the limitations of the existing feedback-based reputation mechanism. In addition, this paper presents a machine learning-based mechanism to identify identifying risky accounts.

Strength:
* Research based on large amounts of real data.
* The findings have a certain contribution to the security of P2P.
* Machine learning-based mechanisms seem reasonable.

Weakness:
* The machine learning methods used in the paper are all basic methods and lack innovation.
* Poor paper writing. The research questions and findings of the study should be listed in the paper to make it easier to read.
* Lack of detailed description of the implementation of machine learning methods.
* Lack of a framework diagram to describe the content and steps of the study.

**Questions:**

Please see the weakness.

**Reviewer Confidence:**

3: The reviewer is confident but not certain that the evaluation is correct

**Scope:**

3: The work is somewhat relevant to the Web and to the track, and is of narrow interest to a sub-community

---

### Official Review · Reviewer_c5Nd · 2023-11-29

**Novelty:** 5
**Technical Quality:** 5

**Review:**

The paper proposes a ML based mechanism to detect suspicious entities in peer-to-peer(P2P) cryptocurrency marketplace.

The paper covers a review on existing P2P cryptocurrency marketplaces to detect the self-promoting attacks possible in online marketplaces and the limitations of feedback-based reputation models in detecting suspicious entity who may launch these attacks. The authors used seven different ML models to detect suspicious activities with a consideration of several external factors.

The paper is well-written, and the application of the proposed model seems novel.

**Questions:**

1.	Can the author clarify what is the significance of contribution number (4) and how the claim is addressed in their work?

2.	Can the author specify why they choose these 7 models, that is, how these 7 are relevant to their application?

**Reviewer Confidence:**

3: The reviewer is confident but not certain that the evaluation is correct

**Scope:**

4: The work is relevant to the Web and to the track, and is of broad interest to the community

---

### Official Review · Reviewer_LRT5 · 2023-12-02

**Novelty:** 4
**Technical Quality:** 4

**Review:**

The paper examines risky vendors in P2P cryptocurrency exchanges from the lens of a 1-year data collection and develops an ML model to predict account suspension.
While I think the first part of the paper examining and exploring the collected data is especially interesting, I found the second part of the paper with ML model exploration extremely lacking. The models used are vanilla models, and none of the insights provided by the paper provide real-world values, given that an adversary can easily adapt and defeat them. Given this fact, I think the authors should focus more on understanding and measuring various artifacts that they touch upon in Sections 3 and 4. The authors could have done much more in understanding quid-pro-quo interactions, measuring white-washing accounts (e.g., new accounts that have similar postings to previously banned ones, or similar IDs, and later would get suspended themselves).

As a side note, I didn't understand the authors' logic for not releasing the dataset (at least to other researchers on a case-by-case basis).

**Questions:**

Given the disparity between the features used in Paxful and LCS, how would a model train on a combination of both datasets perform in the general case? What if you limit the training to the features most highlighted in both models?

**Reviewer Confidence:**

3: The reviewer is confident but not certain that the evaluation is correct

**Scope:**

3: The work is somewhat relevant to the Web and to the track, and is of narrow interest to a sub-community

---

### Decision · Program_Chairs · 2024-01-22

**Decision:**

Accept

**Comment:**

Summary: Reputation systems for P2P cryptocurrency exchanges and ML models to identify risky vendors.


 Strengths:
 + First to study online safety of P2P cryptocurrency marketplaces
 + Current feedback reputation system is ineffective and is manipulable
 + ML method to better predict account suspension
 + Robust and large dataset
 + Some interesting insights


 Weaknesses:
 - ML models used in the study are basic and not novel
 - Insights might be useless against adversaries who are aware of the system used
 - Could be written more clearly
 - Some implementation details missing
 - Concerns about the generality


 Recommendation: The study is interesting and the findings are useful, but mild concerns around the novelty of the ML part of the work.